# The Role of Estrous Synchronization and Artificial Insemination in Improving the Reproductive Performance of *Moo Lath* Gilts

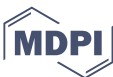

Somsy Xayalath [1,2,*], Gabriella Novotni-Danko [2] and József Rátky [3,*]

[1] Doctoral School of Animal Husbandry, University of Debrecen, Böszörményi Street 138, 4032 Debrecen, Hungary

[2] Institute of Animal Science, Biotechnology and Nature Conservation, Faculty of Agricultural and Food Sciences and Environmental Management, University of Debrecen, Böszörményi Street 138, 4032 Debrecen, Hungary

[3] Department of Obstetrics and Farm Animal Medicine Clinic, University of Veterinary Science, István Street 2, 1078 Budapest, Hungary

* Correspondence: xayalath.somsy@agr.unideb.hu (S.X.); ratky.jozsef@univet.hu (J.R.)

**Abstract:** Considering the different problems facing the Lao indigenous pig breed *Moo Lath*. This study was performed to evaluate the efficiency of applying estrous synchronization for the better reproductive management of this species with the use of Altrenogest Regumate® to increase the litter size and birth weight of crossbred piglets using artificial insemination (AI) with Duroc semen. In total, 36 gilts (age: 6.5−10.5 months, weight at insemination: 36.60−51.42 kg) were used. The gilts were divided into three groups (G1, 2, and 3); G1 (18) were synchronized, while G2 (12) were not. Both G1 and G2 gilts were inseminated using Duroc semen, whereas a local boar naturally serviced the G3 (6) gilts. Our results showed that G1 produced the largest litter, compared with G2 and G3 (8.66, 7.50, and 5.50, respectively; $p < 0.000$). The birth weight of the piglets was not different between the groups ($p = 0.464$), and higher birth weight was observed in the gilts younger than 7 months and lower in those older than 9 months. In conclusion, the litter size of *Moo Lath* primiparous gilts was improved in the F−1 Duroc−*Moo Lath* crossbreed, but their birth weight did not. Moreover, estrous synchronization and AI are novel techniques among Lao farmers who still need more training. Both the optimal body weight and age of gilts at first mating should be clarified for a better economic outcome.

**Keywords:** estrous synchronization; reproductive performance; Lao indigenous pig breeds



## 1. Introduction

In 2020, Laos had a pig population of nearly 4.3 million [1], and almost 91% of them were local breeds [2]. Although the Lao government first imported 120 European pig types from Thailand in early 1980 [3], there are not many commercial pig farms in Laos. Until now, only 578 farms have been registered, with approximately 360,000 pigs in 2019 [4]. The number of commercial pig farms decreased due to the dramatic outbreak of African swine fever (ASF). Since June 2019, Laos has been affected by ASF in all its provinces, including the capital [5], and by other regional diseases with tragic impacts on its small-scale pig production [6]. Nevertheless, there has been a recent increase in the consumption of pork produced from native pigs, particularly in large cities. Presence on local commercial pig farms in Laos is still scarce. Small-scale farmers mainly rear *Moo Lath* pig breeds under the traditional extensive system (low input and productivity). *Moo Lath* sows typically farrow 1 to 1.5 times/year with four to eight piglets each, and few piglets survive at weaning time [7–9]. Between 2016 and 2018, the National Agriculture and Forestry Research Institute (NAFRI) tried to improve the performance of *Moo Lath* sows by crossbreeding with Duroc boar. Nong Taeng black pig was one of the F1 products from

such crossbreeding. Unfortunately, this project ended within two years due to an outbreak (ASF) in the second half of 2019.

Due to the lack of information and training on using assisted reproductive technologies to improve the reproductive performance of *Moo Lath* among Lao farmers, estrous synchronization and artificial insemination (AI) are practically unknown, especially to native pig producers. However, some commercial pig farms in Laos apply AI in their breeding program and use ultrasound to diagnose sow pregnancies between weeks 6 and 7. Therefore, Altrenogest Regumate® treatment and AI were carried out using Duroc semen to improve the reproductive performance of *Moo Lath* pigs, particularly to increase litter size in gilts and the growth performance of piglets. We should also underline that the modern use of native breeds is supposedly the best way of genetic preservation, as it has been achieved by some European pig breeds, i.e., Spanish Iberico and Hungarian Mangalica. In contrast, some Southeast Asian countries lost the majority of their indigenous pig breeds by replacing them with European races.

## 2. Materials and Methods

### 2.1. Location and Duration of Study

The experiment was conducted at the experimental pig farm at Dongkhamxang Agriculture Technical College (DATC) in Vientiane, the capital city of Laos. The warmest month is April, and the coolest is January, while the average maximum temperature around the year is 30 °C, and the minimum is 21 °C. The month with the highest sunshine is December (8.5 h/day), and the lowest one is in August, with about 5 h/day. The rainy season begins in May and ends in September. The dry period is in January, February, November, and December. August is the highest in humidity, while March is the lowest humid month, with around 75% average annual humidity. The annual precipitation is about 1000 mm on average. The experiment was conducted from May 2021 to July 2022, including the preparatory phase.

### 2.2. Animals, Accommodation, and Treatment

The animals used in this study were raised and cared for under the conditions of the Livestock and veterinary Law of Laos (No: 26/NA-Laos, 11 November, 2016 [10]), which covers the protection and animal welfare for all the animals used in this study. Moreover, the experiment aimed to collect only the farm data and, therefore, did not use any blood sample collection or have any objective concerning the animals' bodies. This research project was registered based on the regulation of DATC, with experimental code No.: 054/DATC/2021. According to this regulation, the practices of this study were considered to have a low risk of creating any harm to animals and did not require approval by Lao authorities regarding the care and animals used for scientific purposes. Thirty-six *Moo Lath* gilts (local name of a native pig breed) aged 5−8 months and with a mean body weight of 25−40 kg were used with preparation at the farm. They were kept in individual pens (1.5 × 2 m with a concrete floor) throughout the experimental period. All the gilts were provided with 5 kg rice straw on day 110 of gestation to make the nest. Before insemination, the body weight and backfat thickness of the gilts were recorded, ranging from 36.60 to 51.42 kg and 40.48 to 45.08 mm, respectively. The animals were divided into three groups: G1 ($n$ = 18 gilts) were subject to synchronization using Altrenogest Regumate® (MSD Animal Health, Walton Manor, UK) and inseminated with Duroc semen; G2 ($n$ = 12 gilts) were inseminated with Duroc semen (no synchronization), and G3 ($n$ = 6 gilts) was the control group mated naturally with a mature local boar. It was impossible to find a large number of *Moo Lath* breeding gilts with the same age and same body weight immediately after ASF in Laos. Thus, the main aspect of the selection was to have the necessary number of gilts with cyclic estrous function and approximately similar weight and age.

### 2.3. Feeding Strategies

All the gilts received the same feed and the same portion. Both before and after estrus (flushing phase), each gilt was fed 1.5 kgd$^{-1}$. This amount was reduced to 1.2 kgd$^{-1}$ between days 1 and 28 after insemination. From days 28 to 100 of gestation, each pregnant gilt was fed 1.8 kgd$^{-1}$, and this amount was increased to 2 kgd$^{-1}$ from day 101 of pregnancy until farrowing. During lactation, all the sows received 2.3 kgd$^{-1}$. This feeding strategy is similar to the one described by Middelkoop et al. [11], although the authors in that study used a different breed of pig (Landrace × Large White) from the one used in the current study (*Moo Lath*), which was nearly three times smaller (Figure 1). In this case, the mean body weight of the gilts at the first insemination was 49.59 ± 7.20 kg, and before farrowing, this was 75.99 ± 7.33 kg.

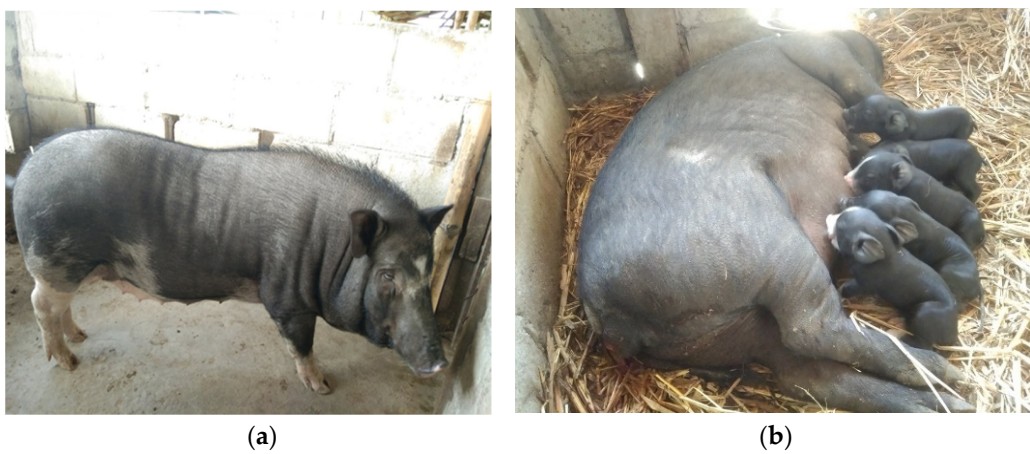

(**a**)　　　　　　　　　　　　　　　　　　　　　(**b**)

**Figure 1.** General appearance of *Moo Lath* gilt (**a**) and sow (**b**) (Photo: Xayalath, 2021 at the experimental farm).

The nutrients of the feed components for the gilts before insemination and gestation are demonstrated in Table 1. All the gilts were fed twice a day at 7:00 h in the morning and 17:00 h in the afternoon. All the gilts had access to clean water *ad libitum*. The dry creep feed was provided as the starter to the piglets on day 7 after farrowing, and all the piglets also had access to clean water ad libitum.

**Table 1.** The nutrient composition of the gilts.

| Nutrient Components | Before Insemination (%) | During Gestation (%) |
|---|---|---|
| DM | 88.50 | 88.36 |
| CP | 15.99 | 15.11 |
| EE | 1.94 | 0.50 |
| CF | 5.26 | 6.22 |
| Ash | 8.48 | 8.18 |
| AIA | 1.40 | 1.69 |

DM: dry matter; CF: crude fiber; CP: crude protein; EE: ether extract; AIA: acid-insoluble ash.

### 2.4. Estrous Synchronization and Artificial Insemination/Mating

All the gilts in G1 were orally fed with 5 mL (20 mg Altrenogest) of Regumate$^®$ at 07:00 h before the main feed in the morning for 18 consecutive days after their first, second, or third estrus. The Regumate$^®$ treatment was according to the MSD Animal Health recommendations and Brüssow et al. [12]. Regumate$^®$ was mixed into a small amount of feed and provided to each gilt in G1. It was monitored and ensured that the gilts received their portion before offering a normal feed. The gilts came into estrus 5–7 days after Altrenogest Regumate$^®$ withdrawal and were inseminated within 6−8 days after the

treatment. All the gilts from G1 and G2 were inseminated three times during their estrus using the same Duroc boar semen, supported by the UDA Pig Farm. AI was performed on the day of estrous detection (in the morning), in the late afternoon, and again on the following morning day. The insemination was performed via boar stimulation in the front corridor. The gilts in control (G3) were also mated three times as described earlier, using a mature native boar (*Moo Lath*) aged 13 months and 80 kg of body weight. Each gilt in G3 was mated using the supervision mating technique, and the boar was taken out from the gilt soon after service.

### 2.5. Pregnancy Determination

Alongside direct observation, none of the gilts returned to the estrous cycle at 21 days after insemination; thus, the gilts were considered pregnant. However, in the current study, pregnancy diagnosis was performed twice for each animal: first on day 28 and another on day 40 after insemination to confirm the pregnancy, using FarmScan® L70 Eye muscles Veterinary Ultrasound as described by [13]. The operation points of the transducer were at the hind abdomen with a 45° angle at the height of the second nipple. A mineral coupling gel application was used for the active transmission of the ultrasound wave.

### 2.6. Farrowing and Weaning Procedures

All the gilts were provided with approximately 5 kg of rice straw at day 110 of pregnancy to prepare a farrowing nest. Each gilt was closely observed and monitored for her farrowing from day 110 until the farrowing day. The piglets were assisted in tearing the membrane of the placenta and dried soon after birth, while the weak piglets were helped when sucking the colostrum (Figure 2). After farrowing, all the piglets had their specific ear-notching mark on days 1−2. All the male piglets were castrated on day 10 after farrowing with the administration of local anesthesia and preventive antibiotics (i.e., a betadine solution and 0.2 mL of kanamycin injection) immediately after the operation. They were also subject to teeth clipping on days 1–2 after farrowing but not tail docking. All the piglets were administered 1 mL of iron injection between 3 and 7 days of age. The piglets were weaned on different days (28 days of age, 35, 42, 49, and 56 days of age) in the afternoon (between 4 and 5 pm) to reduce the stress from the hot weather during the daytime. The weaned piglets were moved to the nursing pens with 8–20 piglets per pen. (Different weaning period is planned for another experiment that will be reported in the near future.)

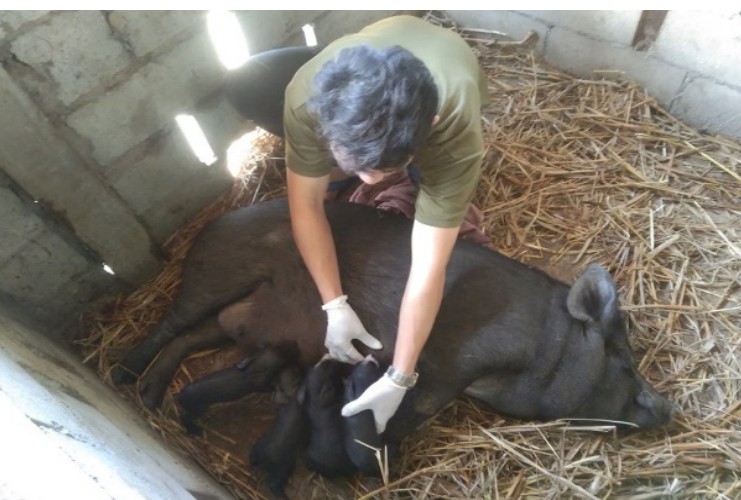

**Figure 2.** Helping the weakened piglets to suck colostrum during the farrowing procedures.

## 2.7. Data Collection and Analysis

The body weight and backfat thickness of each gilt were measured before insemination and on days 70 and 110 of gestation. The duration from the last day of Regumate® feeding to the estrous day of each gilt in G1 (treatment group) was noted. The litter size and livebirth produced by each gilt and the individual body weight of the piglets, including their birth weight (D0), D7, D21, and body weight at day 28 (D28), were recorded. All these data were entered and stored in a Microsoft Excel sheet version 365. The body weight and backfat thickness of the gilts, litter size, live birth, and the piglet weight at D0, D7, D21, and D28 were analyzed using one-way ANOVA and the LSD test of SPSS statistic version 26 (2019) to evaluate the significant differences between the groups. A significant level of 0.05 ($p < 0.05$) was used to indicate a significant difference. The correlation coefficients between the reproductive traits were analyzed using bivariate Pearson. The piglets' growth performance (body weight) was analyzed based on the first 28 days of age to prevent data variation due to different weaning periods.

## 3. Results

### 3.1. Insemination and Pregnancy Diagnosis

All the gilts (in G1) came into estrus within 4 to 7 days after withdrawal of Regumate® feeding, and insemination was completed within 2–3 days of estrous manifestation. In contrast, our team spent nearly two months completing the insemination and mating of the gilts from non-synchronization (G2) and control groups (G3). Out of the 18 gilts in G1, 17 were confirmed to be pregnant on day 28, and 1 gilt was not pregnant until day 40 after insemination and was re-inseminated two days later. All the gilts from non-synchronized and control groups were confirmed to be pregnant at the time of their first diagnosis, performed on day 28 after insemination.

### 3.2. Body Condition of Gilts at Insemination and before Farrowing

There was no difference in the mean age of the gilts before insemination or mating, with approximately $8.17 \pm 1.34$ months. Similarly, the body weight and backfat thickness at insemination and before farrowing did not differ between the groups, with the mean of $49.59 \pm 7.20$ kg and $41.12 \pm 4.77$ mm and $75.99 \pm 7.33$ kg and $40.48 \pm 5.87$ mm, respectively (Table 2). However, the gilts in G3 were found to have a slightly thicker backfat (45.08 mm) at insemination ($p < 0.079$) compared with the gilts in G1 and G2 (40.48 mm and 40.10 mm, respectively).

**Table 2.** Comparison of sows at insemination and before farrowing between groups.

| Parameters | Groups | | | Mean ± SD | *p*-Value | SEM |
|---|---|---|---|---|---|---|
| | G1 | G2 | G3 | | | |
| No. of sows | 18 | 12 | 6 | - | - | - |
| AFI, month | 8.31 | 8.17 | 7.75 | 8.17 ± 1.34 | 0.692 | 0.22 |
| BWI, kg | 49.29 | 51.42 | 46.84 | 49.59 ± 7.20 | 0.446 | 1.12 |
| BFI, mm | 40.48 [a] | 40.10 [a] | 45.08 [b] | 41.12 ± 4.77 | 0.079 | 0.79 |
| BWD110, kg | 77.45 | 74.76 | 74.08 | 75.99 ± 7.33 | 0.498 | 1.22 |
| BFD110, mm | 40.21 | 42.09 | 38.08 | 40.48 ± 5.87 | 0.390 | 0.98 |

[a,b] Means in the same row with different superscripts differ significantly ($p < 0.05$). AFI: age at first insemination; BWI: body weight at insemination; BFI: backfat thickness at insemination; BWD110: body weight day 110; BFD110: backfat thickness day 110 of gestation.

### 3.3. Litter Size and Growth Performance of Piglets

The gilts in the synchronized group (G1) showed a higher mean litter size (8.66, $p < 0.001$) compared with 7.50 in the non-synchronized group (G2) and 5.55 in the control group (G3). In contrast, there were no differences in piglet birth weight ($p = 0.80$) among the

treatment groups. However, the smallest average birth weight was found in G1 (0.62 kg), compared with G2 and G3 (0.65 and 0.66 kg). Stillbirth and piglet mortality before 28 days of age did not differ among the treatment groups. The average piglet mortality within 3 days after farrowing was 0.17 heads (3.0% of the total born in the group) in G3, compared with 0.06 and 0.08 heads in G1 and G2, respectively (Table 3). The number of mummified piglets was higher in the gilts from G2 (0.41 head or 3.21% of the total born in the group) than in G1 (0.22 head or 2.22%) and was not found in the gilts from G3. The mean body weight of the piglets at day 28 of age was significant ($p < 0.001$) between the groups: the piglets of the gilts from G1 had lighter body weight at day 28 after farrowing (3.11 kg) than those in G2 and G3 (3.81 and 3.93 kg, respectively).

**Table 3.** Comparison of the reproductive performance of *Moo Lath* gilts among treatment groups.

| Parameters | Groups | | | Mean ± SD | *p*-Value | SEM |
|---|---|---|---|---|---|---|
| | G1 | G2 | G3 | | | |
| No. of sows | 18 | 12 | 6 | - | - | - |
| LS, head | 8.66 [a] | 7.50 [b] | 5.50 [c] | 7.75 ± 1.33 | 0.001 | 0.22 |
| BW, kg | 0.62 | 0.65 | 0.66 | 0.64 ± 0.15 | 0.464 | 0.01 |
| SB, head | 0.06 | 0.00 | 0.00 | 0.03 ± 0.16 | 0.620 | 0.02 |
| MD3, head | 0.06 | 0.08 | 0.17 | 0.08 ± 0.28 | 0.714 | 0.04 |
| MM, head | 0.22 | 0.41 | 0.00 | 0.25 ± 0.73 | 0.523 | 0.12 |
| BWD28, kg | 3.13 [a] | 3.67 [b] | 3.87 [b] | 3.42 ± 1.08 | 0.001 | 0.09 |

[a,b,c]: Means in the same row with different superscripts differ significantly ($p < 0.05$). LS: litter size; BW: birth weight; SB: stillbirth; MD3: mortality within 3 days after farrowing; MM: mummified; BWD28: body weight at day 28 of age.

The overall growth performance of the piglets from birth to day 28 after farrowing was different ($p < 0.05$) between the groups, especially G1 and G3 (Figure 3). For instance, the mean body weight of the piglets in G1 on day 7 of age (1.33 kg) was 150 g, lighter than piglets in G2 (1.55 kg), and about 210 g in G3 (1.60 kg). This was similar to the mean body weight of the piglets in G1 on day 28 of age (3.13 kg) and was 700 g lesser than the piglets in G2 (3.76 kg) and approximately 820 g lesser than those in G3 (3.87 kg). Furthermore, the average daily gain (ADG) of the piglets in G1 was lower (89 gd$^{-1}$) than the piglets in G2 and G3 (113 and 118 gd$^{-1}$).

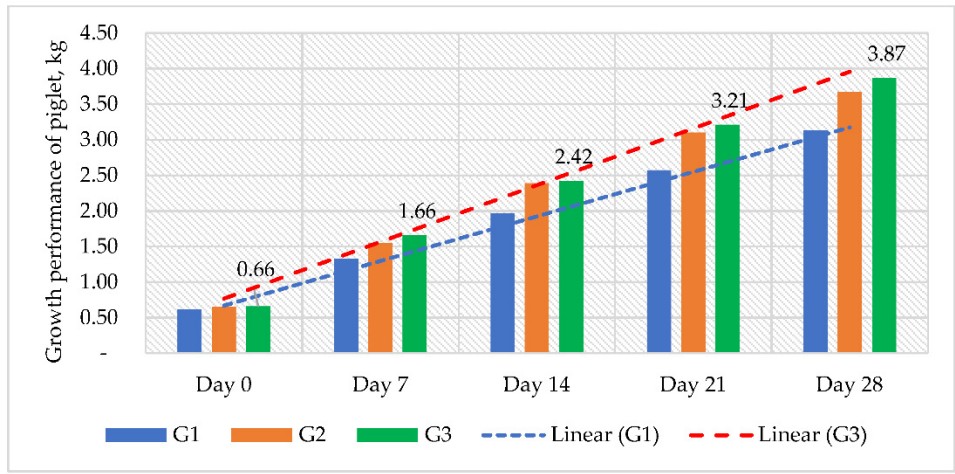

**Figure 3.** Growth performance of piglets among group study from birth to 28 days of age. Day 0, Day 7, Day 14, Day 21, Day 28: body weight of piglet at birth, day 7, day 14, day 21, and day 28 after farrowing.

There was also no difference (*p* = 0.656) in the litter size and birth weight due to the age of the gilts at insemination/mating (Figure 4). However, the smallest litter size (7.50) was found in the sows with less than seven months of age at their first insemination, compared to the sows with ages between 8 and 9 months (8.00) and the sows with ages of over 9 months (7.83) at insemination. The different clustered ages (less than 7 months, between 7 and 9 months, and over 9 months) of the gilts based on their first insemination did not show any difference (*p* = 0.570) in the mean birth weight. Similarly, the different ages (ranging from 6.5 to 10.5 months) at insemination also did not show any differences in stillbirth, mummified piglets, and mortality within 3 days after farrowing.

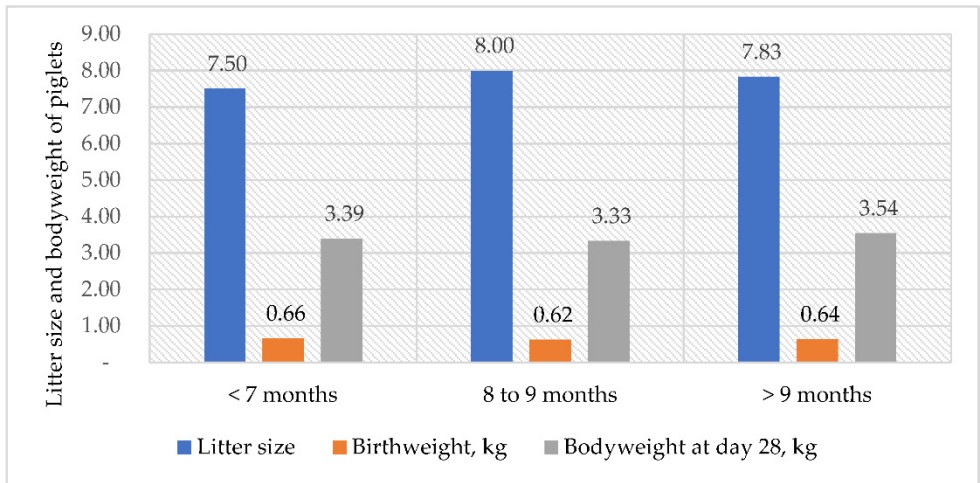

**Figure 4.** Litter size and piglet performance based on the age of gilts at first insemination/mating. <7 months: gilts with less than 7 months of age at insemination; 8 to 9 months: gilts between 8 and 9 months of age at their first insemination; >9 months: gilts aged over 9 months at their first insemination.

Figure 5 shows the body weight of the *Moo Lath* gilts at their first insemination and their reproductive performance, especially their litter size, birth weight, and the growth performance of the piglets on day 28. There was no difference (*p* = 0.799) in the mean litter size based on the body weight of the gilts at the first insemination clustered group. However, larger litter size was found in the sows with body weights of less than 45 kg at their first insemination than the sows with body weights between 45 and 55 kg and those over 55 kg at their first insemination. In contrast, the birth weight of the piglets from the sows with body weights between 45 and 55 kg at their first insemination was lower (0.59 kg) than those of the piglets born from the sows with body weights of less than 45 kg and from the sows with over 55 kg at their first insemination (0.70 kg and 0.62 kg, respectively).

According to Table 4, the age at first insemination (AFI) had a strong positive correlation with body weight at insemination (BWI) and body weight before farrowing (BWF). Similarly, the birth weight (BW) was also correlated to piglet mortality within three days after farrowing (MD3) and the body weight of the piglets on day 28 after farrowing (BWD28). In contrast, both BWI and BWF correlated to the BW of the piglets, and litter size (LS) was estimated to be negatively correlated to the BWD28.

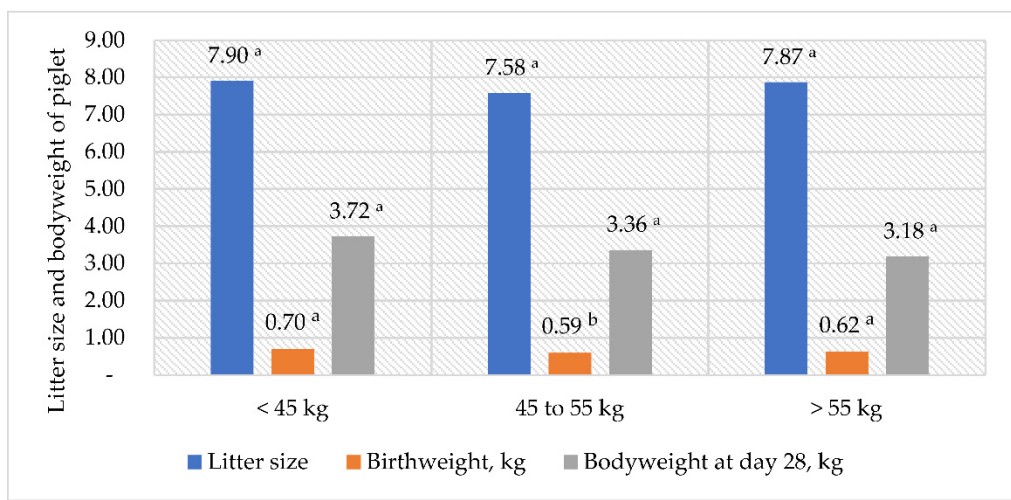

**Figure 5.** Litter size and piglet performance based on body weight of gilts at first insemination. <45 kg: gilts with less than 45 kg of body weight at first insemination; 45 to 55 kg: gilts with between 45 and 55 kg of body weight at first insemination; >55 kg: gilts with over 55 kg of body weight at first insemination; a,b: the mean in the same bar with different superscript is significant difference ($p < 0.05$).

**Table 4.** Correlation coefficients among sows indicate their reproductive performance.

|  | AFI | BWI | BFI | BWF | BFF | LS | BW | WD28 | MM | MD3 |
|---|---|---|---|---|---|---|---|---|---|---|
| AFI |  | 0.507 ** | 0.237 | 0.502 ** | 0.094 | 0.095 | −0.206 | 0.049 | 0.102 | −0.190 |
| BWI | 0.507 ** |  | 0.380 ** | 0.749 ** | 0.094 | −0.064 | −0.360 * | −0.167 | 0.046 | −0.150 |
| BFI | 0.237 | 0.380 * |  | 0.226 | −0.104 | −0.303 | 0.006 | −0.004 | −0.193 | −0.001 |
| BWF | 0.502 ** | 0.749 ** | 0.226 |  | 0.020 | 0.023 | −0.369 * | −0.120 | 0.053 | −0.201 |
| BFF | 0.094 | 0.094 | −0.104 | 0.020 |  | 0.007 | 0.002 | 0.210 | −0.147 | −0.099 |
| LS | 0.095 | −0.064 | −0.303 | 0.023 | 0.007 |  | −0.061 | −0.371 * | 0.211 | −0.095 |
| BW | −0.206 | −0.360 * | 0.006 | −0.369 * | 0.002 | −0.061 |  | 0.478 * | −0.185 | 0.333 * |
| BWD28 | 0.049 | −0.167 | −0.004 | −0.120 | 0.210 | −0.371 * | 0.478 * |  | 0.026 | 0.371 * |
| MM | 0.102 | 0.046 | −0.193 | 0.053 | −0.147 | 0.211 | −0.185 | 0.026 |  | −0.104 |
| MD3 | −0.190 | −0.150 | 0.001 | −0.201 | −0.099 | −0.095 | 0.333 * | 0.371 * | −0.104 |  |

** Correlation is significant at the 0.01 level; * correlation is significant at 0.05 level (2-tailed). AFI: age at first insemination; BWI: body weight at insemination; BFI: backfat thickness at insemination; BWF: body weight before farrowing (day 110); BFF: backfat thickness before farrowing (day 110); LS: litter size; BW: birth weight; MM: mummified; MD3: mortality within 3 days after farrowing; WD28: weaning weight.

## 4. Discussion

The birth weight (BW) of the piglets from G1 and G2 was not higher than that of the piglets from the control group (G3), even though all the gilts of G1 and G2 were inseminated with Duroc semen. We assume that this might be due to the small body weight of the gilts at first insemination, with an average of only 49.59 ± 7.20 kg and 75.99 ± 7.33 kg at farrowing, but they gave higher litter size (7–9 piglets) at farrowing. The slightly higher mean BW of the piglets from G3, 0.66 kg, compared with 0.62 and 0.65 kg in G1 and G2, respectively, is assumed to be due to the smaller litter size in G3 (5.50), compared with those of G1 (8.66) and G2 (7.50). However, further scientific studies might need to confirm the correct response to this issue. Our findings are slightly similar to the findings of Phengvilaysouk et al. [14], who indicated that the mean birth weight (0.6 kg) of *Moo Lath* could not be improved by providing supplementary extra water during gestation and providing enough rice straw to sows 2–3 days before the expected date of farrowing. However, it is quite different from

the average body weight at mating found in our study (72.5 vs. 49.23 kg), around 23.27 kg in difference. In addition, our results are also quite different from the data reported by Keonouchanh [15], who first reported on an improvement in the reproductive performance of Lao native pig breeds due to crossbreeding with Duroc boar. His finding showed the average BW of piglets (local sow × Duroc boar) was 1.16 kg at birth and 6.7 kg on day 30 of age, compared with our finding of 0.64 kg and only 3.42 kg BW on day 28 after farrowing. This might be related to the different body weights at first mating. He used gilts with 80.3 kg of body weight, which is less than the 50 kg gilts used in our study. However, our findings were similar to the results of Manivanh et al. [16], especially regarding the mean body weight before the insemination of the gilts (49.23 vs. 46.30 kg), and the mean litter size (7.41 vs. 7.37). However, our finding regarding the average BW was slightly higher (0.63 vs. 0.51 kg). This might be related to the differences in boar breeds and breeding techniques (Duroc vs. Mong Cai, and mating vs. insemination).

Although the average litter size was different between the studied groups, it is not different from other findings such as 7.37 and 7.8 [14,17], and 7.60 heads [9]. It is quite different from the finding of Wilson [18], who reported the average litter size of *Moo Lath* gilts to be four heads, with about 0.5 kg of birth weight. This difference might be due to the large difference between the body weights at first mating (49.23 vs. 30 kg). In this context, more scientific research is still needed regarding the optimal body weight of *Moo Lath* gilts at their first insemination or mating.

Interestingly, we found three gilts, including one from G1 and two from G2 born with mummified piglets, all occurred in the gilts born with eight and nine litter sizes. It might not be surprising to assume this happened because of the limited capacity of the uterus of *Moo Lath* gilts to enable large litter size, which usually is between four and seven [7,14,17,19]. This hypothesis is similar to that of Cozler et al. [20], who indicated that the increase in litter size has an adverse effect or poses a risk of having mummies in hyper-prolific sows.

Another interesting point to note is that the age at first insemination did not influence the litter size and birth weight between the clustered age groups. However, the gilts with less than 7 months of age at insemination had the smallest ($p > 0.05$) litter size of 7.50, compared with 8.00 in the gilts with ages between 8 and 9 months, and about 7.80 in the gilts with the ages of over 9 months at their first insemination or mating. It is assumed that this is due to the earlier than six months of onset puberty in *Moo Lath* gilts [18]. According to the present findings regarding litter sizes among the cluster groups (Figure 4), it was considered that the optimal age of *Moo Lath* gilts for their first insemination/mating could be between 7 and 8 months, instead of over 8 months of age. Due to the limitation of the supported data, more in-depth scientific research is needed about the optimal ages of *Moo Lath* gilts for their first insemination/mating, which might affect their reproductive performance. In particular, piglet quality and the productive life of sows might be affected by their different levels of reproductive capacity. For instance, Silva et al. [21] reported that indigenous pigs in Vietnam produced their first farrowing at 8.40 months of age, while in Sri Lanka, this was 6.4 months.

The optimal body weight of *Moo Lath* gilts is also another point that is worth noting in this study. As shown in Figure 5, the gilts with less than 45 kg at their first insemination had the largest litter size (7.90), compared with the sows between 45 and 55 kg (7.58) and those more than 55 kg (7.87). Similarly, the birth weight and growth performance of the piglets at day 28 from the sows with body weights of less than 45 kg were larger than those from the rest of the two clustered groups. From these results, it could be seen that the optimal body weight of *Moo Lath* gilts should be a bit less than 45 kg. This hypothesis supports the findings of Wilson [18] and Keonouchanh et al. [19], who reported that the body weight of *Moo Lath* gilts at first mating was 30 kg and 39 kg, respectively.

This study covered 279 born piglets from experimental animals, including 51% female and 49% male piglets. No difference was found in the mean birth weight ($p = 0.298$) between the male (0.64 kg) and female (0.63 kg) piglets. Our findings are in opposition to those of Keonouchanh [15], who reported that the ratio of the male- and female-born piglets of the

Back Nong Taeng (Lao Local sow × Duroc boar) was 40% vs. 60%, a difference of 20%. Both studies' experimental breeding conditions were very similar; only breeding techniques differed (insemination vs. mating). It might be quite difficult to explain the different results in this case. A difference in the gender ratio was reported by Bocian et al. [22], who found that primiparous sows gave birth to 9.8% more male piglets. They also showed that female piglets were heavier in both birth weight and weaning weight than males (1.35 vs. 1.23 kg, and 6.90 vs. 6.68 kg, respectively). It was assumed that these opposites might be attributed to the breeds' differences (*Moo Lath* sows × Duroc boar vs. Polish Large White sows × Polish Landrace boars). Regarding the performance of *Moo Lath* gilts and sows, there is a lack of research that addresses the sex ratio of the piglets. Therefore, more scientific research on piglet performance related to sex differences is needed.

## 5. Conclusions

Using Altrenogest Regumate® and artificial insemination, the litter size of *Moo Lath* primiparous sows can be improved in the F-1 Duroc–*Moo Lath* crossbred, as the gilts in G1 gilts had larger litter sizes by 1.16 piglets than those in G2 and by 3.16 piglets than those in G3. However, the piglets' birth weight and growth performance on day 28 cannot be improved in F-1. The age at first insemination did not influence the reproductive performance of *Moo Lath* gilts in the present study. The optimal age at the first service of *Moo Lath* is between 7 and 8 months. For their first mating, *Moo Lath* gilts should weigh slightly less than 45 kg, which is the typical situation. However, future studies are required to clarify the optimal body weight influence on the reproductive performance of *Moo Lath* gilts, particularly their litter size and birth weight. Estrous synchronization and AI are very new for pig producers in Laos, especially small-scale farmers, and more scientific studies are needed, especially on indigenous pig breeds.

**Author Contributions:** Both authors (S.X. and J.R.) equally contributed to designing the experiment, carrying out the research, and preparing the manuscript. G.N.-D. helped to provide advice on the technical aspect and helped to revise the manuscript. All authors have read and agreed to the published version of the manuscript.

**Funding:** Spendium Hungaricum Programme, Dongkhamxang Agriculture Technical College, VITAFORT Co., in Laos, and UDA Import-Export Public Company supported this work.

**Institutional Review Board Statement:** Not applicable.

**Informed Consent Statement:** Not applicable.

**Data Availability Statement:** The datasets are available upon request from the corresponding author.

**Acknowledgments:** The author acknowledge the Stipendium Hungaricum Scholarship Program, which provided me with the opportunity to study in Hungary. The authors acknowledge all corning teams from Dongkhamxang Agriculture Technical College (DATS), Laos, who strived hard to carry out the experiment. Similarly, the authors also acknowledge UDA FARM, who supported us with the Duroc semen that was applied in the experiment. Last but not least, the authors also acknowledge the VITAFORT Co., in Laos, who supported the feed for the experiment.

**Conflicts of Interest:** The authors declare no conflict of interest.

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
