# Peer review of "The Role of Estrous Synchronization and Artificial Insemination in Improving the Reproductive Performance of Moo Lath Gilts"

_agriculture, doi:10.3390/agriculture12101549_

Round 1

Reviewer 1 Report

Following recommendations are made after review to improve the manuscript

Abstract: Please clearly indicate what Regumate is. Also, rephrase the sentences in the attached file to clarify the study groups.

The introduction is well written. Materials and methods: Again, clearly indicate which drug is present in Regumte. Improve the sentence expression where authors are quoted for a procedure.

Results: Improve Figure 3. Include the G3 group as a bar instead of a line. Also, if the results are statistically non-significant between the groups, then authors should not further describe the results to indicate mere numerical differences. Keeping this in view, please improve the result section. 

Conclusion: Conclusions need modification. Authors should only describe what is proven through the results of the study. Remove the sentences which appear irrelevant to the results. Future direction or suggestion may be discussed separately. For specific comment please see the attached file.

Author Response

September 21, 2022

Dear reviewer   

Thank you very much for your reviewing our manuscript entitled "The Role of Estrous Synchronization and Artificial Insemination in Improving Reproductive Performance of Moo Lath Gilts". We appreciate all your efforts in reviewing our manuscript. We are grateful for all your valuable comments and suggestions to improve our manuscript, which we humbly accept. We have carefully revised and responded to all points of your comments, and we have highlighted what we have improved as follows:

Abstract:

Comment 1: Line 14 [please indicate the active drug first then the commercial name]

Response: We added: [Altrenogest Regumate®]

Comment 2: Line [Rephrase the sentences: Gilts were divided into 3 groups, inseminated with Duroc semen, G1 (18 16 gilts) were synchronized while G2 (12 gilts) were not. G3 (6 gilts = control group) were bred natu-17 rally with a mature local boar.]

Response: We improved the sentences: [Gilts were divided into 3 groups(G1, 2, and 3); G1 (18) were synchronized, while G2 (12) were not. Both G1 and G2 gilts were inseminated using Duroc semen, whereas a local boar naturally serviced G3 (6) gilts.]

Comment 3: Line 22 [replace cannot with did not]

Response: we agreed to replace “cannot” with did not

Introduction:

Comment 1: Line 50 [Indicate the active drug followed by the commercial name: Regumate® treatment]

Response: We added “Altrenogest in front of Regumate: [Altrenogest Regumate® treatment].

Materials:

Comment 1: Line 95 [improve the sentence expression as it is not clear who was no. 11: This feeding strategy is similar to that of…]

Response: We improved the whole sentence: [This feeding strategy is similar to the one described by Middelkoop et al. [11], although the authors used a different breed of pig (Landrace x Large White) from the one used in the current study (Moo Lath), which was nearly three times smaller].  

Comment 2: Line 111 [again indicate the active drug fed to the animals: Regumate®]

Response: We modified the phrase: [All gilts in G1 were orally fed with 5 ml (20 mg Altrenogest) of Regumate® at 07:00 h….].

Comment 3: Line 113 [Please improve the writing style: conducted as described by]

Response: We improved the sentence: [The Regumate® treatment was according to the MSD Animal Health recommendations and Brüssow et al.].

Results:

Comment 1: Line 203 [add comma afterward: …piglets in G1 on day 7 of age (1.33 kg) was 150 g …]

Response: We added “,” after 150 g: […piglets in G1 on day 7 of age (1.33 kg) was 150 g, lighter than….].

Comment 2:[ In figure 3: Add G3 group as bar rather than a line]

Response: We agreed to change the line to a bar for G3, and we kept two linear of G1 and G2 for comparison.  

Comment 3: Line 213-215 [In conclusion author argue that farmer should breed Moo lath pig at early age however the study indicates that breeding at early age results in smaller litter size; please explain this point in discussion and accordingly change the conclusion: However, the smallest litter size (7.50) was found in the sows with less than seven months of age at their first insemination, compared to the sows with ages between 8 to 9 months (8.00)]

Response: We agreed to give more explanations in the discus and improve the conclusion. But please note that our recommendation for Lao farmers is that the gilts should be between 7 and 8 months. This means it should not be less than 7 months of age. The idea is to (1) save the cost for gilts’ development; (2) based on our findings, the peak litter size of Moo Lath gilts for their first service might be 8 months of age. So, Lao native pig producers should not wait until their gilts reach to be older than 8 months of age. 

Comment 4: Line 228-230 [when there is no statistical difference for the litter size based on age then it is no necessary to indicate numerical differences; delete it or explain it in discussion: [However, a larger litter size (7.90) was found in sows with bodyweight at first insemination less than 45 kg, compared to 7.58 in sows with bodyweight between 45 to 55 kg, and 7.87 in sows with over 55 kg at their first insemination.]

Response: We agreed to improve the sentence: [However, a larger litter size was found in sows with bodyweight less than 45 kg at their first insemination, compared to sows with bodyweight between 45 to 55 kg, and sows with over 55 kg at their first insemination].  And, it had been more discussed in the discussion.

Conclusion:

Comment 1: [Conclusions need modification. Authors should only describe what is proven through the results of the study. Remove the sentences which appear irrelevant to the results. Future direction or suggestion may be discussed separately. For specific comment please see the attached file.]

Response: Thank you for pointing out this unclear point, and we decided to modify the conclusion: [Using Altrenogest Regumate® and artificial insemination, the litter size of Moo Lath primiparous sows can be improved in F-1 Duroc-Moo Lath crossbred as gilts in G1 gilts have larger litter sizes by 1.16 piglets than G2 and 3.16 piglets than the G3. However, the piglets’ birth weight and growth performance at day 28 cannot be improved in F-1. Age at first insemination did not influence the reproductive performance of Moo Lath gilts in the present study. The optimal age at first service of Moo Lath is between 7 and 8 months. For their first mating, Moo Lath gilts should weigh slightly less than 45 kg, which is the typical situation. However, future studies are required to clarify the optimal bodyweight influence on reproductive performance, particularly litter size and birth weight. Estrus synchronization and AI are very new for pig producers in Laos, especially small-scale farmers, and it needs more scientific studies, especially on indigenous pig breeds.]

Other improvements:

  • We added a new table (table 1) for nutrient components of feed, which will be better than sentences.
  • We separated table 1 (old) into (table 2) and (table 3) to make them clear and separate between body conditions of sows and piglets’ performance.

We also carefully checked all spelling and grammatical errors and corrected them as necessary.

Thank you so much for considering this manuscript.

Best regards,

All behalf of all-co-authors

Reviewer 2 Report

A very interesting work concerning the breeding performance of native breeds of sows. The topic is current in many countries, because more and more attention is paid to native breeds, their utility, not only for reproduction, but also for fattening and slaughter. The subject matter raised by the authors is therefore very topical as it is ultimately related to the profitability of pig keeping, which is drastically falling, which is a problem in many countries. The use of local breeds of pigs that are resistant to often harsh environmental conditions (such as in Lao) is a solution in combination with the use of breeds that undergo intensive breeding work. By adding artificial insemination techniques to reproduction, it is possible to achieve increased fertility results, and this may allow an increase in the number of pigs in a specyfic area.

The form of citation - methodology line 95-96 - one should write the name according to whose method the feeding was used or change the order of the sentence (e.g. "This feeding strategy is similar to that of [11]" - suggested change of the order of the sentence or writing according to whose method (surname and then the reference number.) This comment also applies to a number of citations occurring, among others, in the discussion. I suggest that the form of writing the citations in the text should be changed.

Lines 98-101 - abbreviations concerning the percentages of individual feed components in the ration are not explained. It would look good to write the ingredients in the table and the individual abbreviations below it.

Table 1 - it is unnecessary to write the significance symbol next to numerical values, if there were no statistically significant differences between the features.

Table 1 - I suggest separating or listing the piglet results in a separate table. They will then be more readable.

Author Response

September 21, 2022

Dear reviewer   

Thank you very much for your reviewing our manuscript entitled "The Role of Estrous Synchronization and Artificial Insemination in Improving Reproductive Performance of Moo Lath Gilts". We appreciate all your efforts in reviewing our manuscript. We are grateful for all your valuable comments and suggestions to improve our manuscript, which we humbly accept. Your comments are very favorable for our work. We have carefully revised and responded to all points of your comments, and we have highlighted what we have improved as follows:

Comment 1: [The form of citation - methodology line 95-96 - one should write the name according to whose method the feeding was used or change the order of the sentence (e.g. "This feeding strategy is similar to that of [11]" - suggested change of the order of the sentence or writing according to whose method (surname and then the reference number.) This comment also applies to a number of citations occurring, among others, in the discussion. I suggest that the form of writing the citations in the text should be changed.]

Response: We improved the whole sentence: [This feeding strategy is similar to the one described by Middelkoop et al. [11], although the authors used a different breed of pig (Landrace x Large White) from the one used in the current study (Moo Lath), which was nearly three times smaller]. And, we also agreed to improve all unclear citations in the discussion.

Comment 2: [Lines 98-101 - abbreviations concerning the percentages of individual feed components in the ration are not explained. It would look good to write the ingredients in the table and the individual abbreviations below it.]

Response: Thank you for your raising points. So, we decided to insert table:

  • Table 1. The nutrient composition of the gilts

Comment 3: [Table 1 - it is unnecessary to write the significance symbol next to numerical values, if there were no statistically significant differences between the features.]

Response: We agreed to remove all superscripts from the numerical values in which p > 0.05, or no significant differences.

Comment 3: [Table 1 - I suggest separating or listing the piglet results in a separate table. They will then be more readable.]

Response: We agreed to separate (Table 1) into two tables:

  • Table 2. Comparison of sows at insemination and before farrowing between groups; and
  • Table 3: Comparison of the reproductive performance of Moo Lath gilts among treatment groups

Other improvements:

  • We modified the Figure 3 by changing the line of G3 to a bar, and we kept two linear of G1 and G2 for comparison.
  • We modified the conclusion to make it more specific: [Using Altrenogest Regumate® and artificial insemination, the litter size of Moo Lath primiparous sows can be improved in F-1 Duroc-Moo Lath crossbred as gilts in G1 gilts have larger litter sizes by 1.16 piglets than G2 and 3.16 piglets than the G3. However, the piglets’ birth weight and growth performance at day 28 cannot be improved in F-1. Age at first insemination did not influence the reproductive performance of Moo Lath gilts in the present study. The optimal age at first service of Moo Lath is between 7 and 8 months. For their first mating, Moo Lath gilts should weigh slightly less than 45 kg, which is the typical situation. However, future studies are required to clarify the optimal bodyweight influence on reproductive performance, particularly litter size and birth weight. Estrus synchronization and AI are very new for pig producers in Laos, especially small-scale farmers, and it needs more scientific studies, especially on indigenous pig breeds.]

We also carefully checked all spelling and grammatical errors and corrected them as necessary.

Thank you so much for considering this manuscript.

Best regards,

All behalf of all-co-authors
